# A Simple and Efficient Method for RSS-AOA-Based Localization with Heterogeneous Anchor Nodes

**DOI:** 10.3390/s25072028

**Published:** 2025-03-24

**Authors:** Weizhong Ding, Lincan Li, Shengming Chang

**Affiliations:** School of Cyber Science and Engineering, Ningbo University of Technology, Ningbo 315211, China; weizhongdingnubt@163.com (W.D.); csm20130504@163.com (S.C.)

**Keywords:** localization, received signal strength, angle of arrival, linear-weighted least squares

## Abstract

Accurate and reliable localization is crucial for various wireless communication applications. A multitude of studies have presented accurate localization methods using hybrid received signal strength (RSS) and angle of arrival (AOA) measurements. However, these studies typically assume identical measurement noise distributions for different anchor nodes, which may not accurately reflect real-world scenarios with varying noise distributions. In this paper, we propose a simple and efficient localization method based on hybrid RSS-AOA measurements that accounts for the varying measurement noises of different anchor nodes. We develop a closed-form estimator for the target location employing the linear-weighted least squares (LWLS) algorithm, where the weight of each LWLS equation is the inverse of its residual variance. Due to the unknown variances of LWLS equation residuals, we employ a two-stage LWLS method for estimation. The proposed method is computationally efficient, adaptable to different types of wireless communication systems and environments, and provides more accurate and reliable localization results compared to existing RSS-AOA localization techniques. Additionally, we derive the Cramer–Rao lower bound (CRLB) for the RSS-AOA signal sequences used in the proposed method. Simulation results demonstrate the superiority of the proposed method.

## 1. Introduction

With the rapid development and widespread utilization of wireless sensor networks (WSNs) [1,2,3,4,5] and mobile devices, the demand for accurate and reliable localization has increased significantly. This is particularly evident in indoor positioning [6], autonomous navigation [7], and intelligent transportation systems [8], where the ability to locate objects, vehicles, and people with high accuracy is essential. Additionally, in the context of the Internet of Things (IoT) [9], localizing devices and objects has become increasingly vital for optimizing IoT system performance, reducing energy consumption, and improving user experiences. Consequently, the creation of localization techniques that are accurate, reliable, and efficient has emerged as an essential research focus in recent years. One of the most promising localization approaches involves the use of hybrid received signal strength (RSS) and angle of arrival (AOA) measurements. By merging these two signals, it is possible to achieve greater accuracy and robustness than by relying on either signal independently.

Hybrid RSS-AOA measurements have been widely used for localization. The measurement model utilizes RSS measurements to represent distance (where the RSS measurement value decreases as the distance increases), and AOA measurements to represent angle. This approach allows three-dimensional localization to be achieved with only one anchor node whose location is known, while using a single measurement model requires three or more anchor nodes. Additionally, the RSS-AOA measurement model offers high robustness against shadowing and multipath effects [10], which makes it a more resilient option compared to other localization methods.

Numerous localization methods based on hybrid RSS-AOA measurements have been proposed. Among them, linearly weighted least squares (LWLS) methods are promising and most widely used, offering both high accuracy and low computational complexity. The main procedure of this approach involves converting the distance between anchor nodes and the target node into a linear expression of the target node’s location using spherical coordinate transformation. The LWLS method is then utilized to construct a closed-form solution for the target node’s location.

The accuracy of these methods depends on the choice of weight matrices, and various weight matrix computation methods have been proposed in the literature to achieve high localization accuracy.

Many commonly used weight matrix computation methods have been proposed in the literature. The least squares (LS) method in [11], which did not use any weight matrix (i.e., the weight matrix was a unit matrix), was often used as a benchmark for comparison in simulations. The distance-related WLS method, as proposed by Tomic et al. in [12], has been shown to improve localization accuracy without adding complexity. However, using weights that only change based on the distance between the target node and anchor nodes may not be optimal. Kang et al. [13] developed the error covariance-weighted least squares (ECWLS) method, which computes weights based on an approximate error covariance matrix calculated from the LS method estimation of the target location. This approach led to more accurate weights, but estimating the variance of measurement noise was challenging, and directly multiplying the estimated variance into the weight could increase error. Watanabe presented the two-step error variance-weighted least squares (TSLS) method based on AOA measurement [14], which estimated the target location using the LS method and then calculated the variance as the weight. However, this method only considered the noise variance of the evaluation function items and not the noise value of the measurements. To address this, Ding et al. [15] proposed an error variance and noise value WLS (ENWLS) that could adjust the weight to account for measurement noise values, giving more weight to measurements with smaller noise values.

In addition to the LWLS methods, two other commonly used approaches were convex relaxation [16] and constructing the generalized trust-region subproblem (GTRS) [17]. The main process of using convex relaxation for localization involved formulating a convex optimization problem that sought to minimize the localization error subject to a set of constraints. Convex relaxation methods could achieve high-precision localization, but they were computationally complex and therefore not suitable for real-time applications. The main process of constructing GTRS for localization involved formulating the GTRS and using the bisection method to search for the optimal solution. To provide a benchmark for comparison in simulations, this paper included two state-of-the-art methods based on convex relaxation and GTRS as references. The SDP-SOCP method was proposed by Chang et al. [18] and had been shown to achieve high accuracy. To formulate the convex optimization problem, semi-definite programming (SDP) and second-order cone programming (SOCP) were used to transform the non-convex system into a convex system. However, the complexity of these methods was relatively high, which might limit the practical applications. In [19], Tomic et al. proposed a novel localization method based on GTRS, which derived a non-convex estimator that could be transformed into a GTRS framework.

However, all the above-mentioned RSS-AOA localization methods assume that the measurement noise of all the anchor nodes follows the same distribution, which does not accurately reflect the real-world scenarios. In reality, measurement noise distributions can vary significantly due to factors such as device type, node location, antenna orientation, and signal propagation environment. Therefore, while these methods can achieve high accuracy in simulations, they may have significant localization errors in practical scenarios.

To address this issue, this paper proposes a novel approach that takes into account the varying measurement noise distributions of different nodes and estimates the weights of each LWLS equation based on its residual variance. However, the variances of LWLS equation residuals are unknown, and thus we propose a two-stage LWLS method to estimate the variance of each LWLS equation. In the first stage, we use the LWLS method based on the distances between the target node and the anchor nodes to calculate a rough location estimate. In the second stage, we use the rough estimate as input to the LWLS estimator to calculate the residual for each equation, and then use the inverse of the residual variances as the weights to calculate an accurate estimate of the target node’s location using LWLS.

Calculating the variance of each LWLS equation requires a time series of measurements, which is referred to as the RSS-AOA time series in this paper. The RSS-AOA time series refers to a series of mixed RSS-AOA measurements arranged in chronological order. RSS time series are often utilized to improve the positioning accuracy in fingerprint-based localization methods [20,21,22,23]. The model of RSS-AOA time series is introduced in detail in Section 2.

To demonstrate the excellent localization performance of the proposed method, this paper compared the proposed method with the state-of-the-art methods in simulation.

In localization, Cramer–Rao lower bound (CRLB) is a commonly used performance lower bound for evaluating the performance of localization methods. CRLB describes the minimum variance of estimated location error, which means that no matter what estimation method is used, the estimated location error variance cannot be less than this lower bound. By comparing the performance of algorithms with CRLB, the feasibility and reliability of methods can be determined, and whether the methods have reached the theoretically optimal performance can also be determined. The CRLB of the RSS-AOA time series model is different from that of the traditional hybrid RSS-AOA model. In Section 2, the CRLB of the RSS-AOA time series model is derived.

Through simulation experiments, we found that the proposed method has good performance in different positioning scenarios and is very close to the CRLB. Although the computational complexity is slightly higher than that of the traditional LWLS (higher computational complexity when calculating weights), the computational complexity is still within an acceptable range, and the calculation speed is fast, which can achieve real-time positioning. This indicates that the proposed method can achieve high accuracy, reliability, and real-time performance.

The main contributions of this paper are as follows:(1)Diverse Anchor Node Scenarios: This paper considers a more practical scenario, specifically localization scenarios involving different types of anchor nodes, which closely reflects real-world situations.(2)Innovative Algorithm: A novel localization method based on hybrid RSS-AOA measurements is proposed, which accounts for the differences in measurement noise distributions across different nodes to enhance localization accuracy and reliability.(3)Two-stage LWLS Method: This paper introduces a two-stage LWLS method for estimating the residual variance of each LWLS equation, thereby achieving more accurate target node localization.(4)RSS-AOA Time Series Model: This paper incorporates an RSS-AOA time series model that utilizes the time series information of RSS-AOA measurements to further improve localization accuracy.(5)CRLB Derivation: A detailed derivation of the CRLB of the RSS-AOA time series model is provided, establishing a theoretical foundation for evaluating the performance of localization methods.

In the remaining sections of this paper, the sequential hybrid RSS-AOA system model is presented and the CRLB of the measurement model is derived in Section 2. The details of the proposed method are shown in Section 3. The extensive simulation results demonstrate the good performance of the proposed method in Section 4. We conclude the paper with a summary and discussion of the future work in Section 5.

## 2. Problem Formulation and Cramer–Rao Lower Bound

In this section, the system model is first described in detail, followed by the formulation of the localization problem. The CRLB for RSS-AOA time series-based localization is also derived.

### 2.1. System Model and Problem Formulation

Consider a three-dimensional (3D) WSN consisting of *N* anchor nodes whose locations are known, denoted by $a_{i}={\left[a_{i1},a_{i2},a_{i3}\right]}^{T},i=1,\cdots ,N$, and a target node whose location is unknown, denoted by $x={\left[x_{1},x_{2},x_{3}\right]}^{T}$.

As shown in Figure 1, the true value of the azimuth angle and elevation angle between the target node and the *i*-th anchor node is referred to as ${\dot{\phi }}_{i}$ and ${\dot{\alpha }}_{i}$, which can be written as(1)$$\begin{array}{c} {\dot{\phi }}_{i}=\arctan\left(\frac{x_{2}-a_{i2}}{x_{1}-a_{i1}}\right),\;i=1,\cdots ,N, \end{array}$$(2)$$\begin{array}{c} {\dot{\alpha }}_{i}=\arccos\left(\frac{x_{3}-a_{i3}}{∥x-a_{i}∥}\right),\;i=1,\cdots ,N, \end{array}$$
where ${\dot{\phi }}_{i}\in \left(-\pi ,\pi \right)$ and ${\dot{\alpha }}_{i}\in \left(-\frac{\pi }{2},\frac{\pi }{2}\right)$.

The RSS measurements without noise are denoted by(3)$$\begin{array}{c} {\dot{P}}_{i}=P_{0}-10\gamma \log_{10}\frac{∥x-a_{i}∥}{d_{0}},\;i=1,\cdots ,N, \end{array}$$
where $P_{0}$ represents the reference received signal power when the reference distance is $d_{0}$, and γ is referred to as the path loss exponent which is environment-related.

Taking measurement noise into consideration, the hybrid RSS-AOA time series model can be represented as(4)$$\begin{array}{c} \phi_{it}={\dot{\phi }}_{i}+m_{it},\;t=1,\cdots ,T, \end{array}$$(5)$$\begin{array}{c} \alpha_{it}={\dot{\alpha }}_{i}+v_{it},\;t=1,\cdots ,T, \end{array}$$(6)$$\begin{array}{c} P_{it}={\dot{P}}_{i}+n_{it},\;t=1,\cdots ,T, \end{array}$$
where *T* denotes the number of time steps, which means the total number of discrete time intervals or steps in the time series. $m_{it}$, $v_{it}$, and $n_{it}$ refer to measurement noise of the *t*-th time step of the azimuth angle, elevation angle and received signal strength, respectively, between the target node and the *i*-th anchor node. Generally, it is assumed that $\left\{m_{it}\right\}$ follows independent identical distributions, where i=1,⋯,N and t=1,⋯,T. $\left\{v_{it}\right\}$ and $\left\{n_{it}\right\}$ also follow independent identical distributions. However, AOA and RSS measurements are subject to measurement noise that is related to many factors, such as the quality of the hardware components, the location of the nodes, the orientation of the antennas, and the characteristics of the signal propagation environment. Therefore, it is possible that the variances of measurement noises from different anchor nodes are different. So, in this paper, the measurement noise from different anchor nodes follows different distributions, which can be modeled as a zero-mean Gaussian random variables, i.e., $m_{it}∼N\left(0,\sigma_{m_{i}}^{2}\right)$, $v_{it}∼N\left(0,\sigma_{v_{i}}^{2}\right)$, $n_{it}∼N\left(0,\sigma_{n_{i}}^{2}\right)$, where $\sigma_{m_{i}}$, $\sigma_{v_{i}}$ and $\sigma_{n_{i}}$ are the standard deviation of $m_{it}$, $v_{it}$ and $n_{it}$, respectively. To make the standard deviation of measurement noise of different anchor nodes different, such that $m_{p}$ is not equal to $m_{q}$, p≠q, *p*, q=1,⋯,N, the standard deviations can be assumed to follow exponential distributions with the means of $\mu_{m}$, $\mu_{v}$, and $\mu_{n}$. It can be expressed as $\sigma_{m_{i}}∼\mathrm{Exp}\left(\mu_{m}\right)$, $\sigma_{v_{i}}∼\mathrm{Exp}\left(\mu_{v}\right)$, and $\sigma_{n_{i}}∼\mathrm{Exp}\left(\mu_{n}\right)$. Then, the conditional probability density function (PDF) of the hybrid RSS-AOA measurement θ that x is given can be expressed as(7)$$\begin{array}{c} \begin{array}{c} P\left(\theta |x\right)=\prod_{i=1}^{N}\prod_{t=1}^{T}\left(\frac{1}{\sqrt{2\pi \sigma_{m_{i}}}}\exp\left\{-\frac{{\left(\phi_{it}-\arctan\left(\frac{x_{2}-a_{i2}}{x_{1}-a_{i1}}\right)\right)}^{2}}{2\sigma_{m_{i}}^{2}}\right\}\right.\\ \left.\cdot \frac{1}{\sqrt{2\pi \sigma_{v_{i}}}}\exp\left\{-\frac{{\left(\alpha_{it}-\arccos\left(\frac{x_{3}-a_{i3}}{∥x-a_{i}∥}\right)\right)}^{2}}{2\sigma_{v_{i}}^{2}}\right\}\right.\\ \left.\cdot \frac{1}{\sqrt{2\pi \sigma_{n_{i}}}}\exp\left\{-\frac{{\left(P_{it}-P_{0}+10\gamma \log_{10}\frac{∥x-a_{i}∥}{d_{0}}\right)}^{2}}{2\sigma_{v_{i}}^{2}}\right\}\right), \end{array} \end{array}$$
where $\theta ={\left[\phi_{11},\phi_{12},\cdots ,\phi_{1T},\phi_{N1}\cdots ,\phi_{NT},\alpha_{11},\alpha_{12},\cdots ,\alpha_{1T},\alpha_{N1}\cdots ,\alpha_{NT},P_{11},P_{12},\cdots ,P_{1T},P_{N1},\cdots ,P_{NT}\right]}^{T}$. It is worth noting that modeling the standard deviations of different anchor nodes in real-world localization scenarios is challenging, and Gaussian models are a possible and reasonable approach. However, in this paper, exponential distributions are considered to account for the possibility of extremely imprecise anchor nodes, allowing for greater variation in the precision of different nodes.

Then, the maximum likelihood (ML) estimator of the hybrid RSS-AOA measurement is obtained as(8)$$\begin{array}{c} \hat{x}=\arg\max_{x}P\left(\theta |x\right). \end{array}$$

After derivation, the ML estimation can be expressed as(9)$$\begin{array}{c} \begin{array}{cc} \hat{x}=\arg\min_{x}\sum_{i=1}^{N}\sum_{t=1}^{T}( & \frac{{\left(\phi_{it}-\arctan\left(\frac{x_{2}-a_{i2}}{x_{1}-a_{i1}}\right)\right)}^{2}}{\sigma_{m_{i}}^{2}}\\ \; & +\frac{{\left(\alpha_{it}-\arccos\left(\frac{x_{3}-a_{i3}}{∥x-a_{i}∥}\right)\right)}^{2}}{\sigma_{v_{i}}^{2}}\\ \; & +\frac{{\left(P_{it}-P_{0}-10\gamma \log_{10}\frac{∥x-a_{i}∥}{d_{0}}\right)}^{2}}{\sigma_{n_{i}}^{2}}). \end{array} \end{array}$$

### 2.2. Cramer–Rao Lower Bound

The Cramer–Rao lower bound (CRLB) is a theoretical lower bound on the variance of any unbiased estimator of a parameter in a statistical model. In other words, it provides a lower limit on how well we can estimate a parameter, given the statistical properties of the data and the model used to describe them. In the hybrid RSS-AOA measurement sequences, the CRLB of location estimation depends on the locations of the anchor nodes, the true location of the target node, and the measurement noise variances.

The CRLB can be calculated based on the Fisher information matrix (FIM). The FIM is a mathematical tool that measures the amount of information that an observable random variable carries about an unknown parameter in a statistical model. In the localization system model, it can be defined as(10)$$\begin{array}{c} {\left[\mathrm{FIM}(x)\right]}_{ij}=-E\left[\frac{\partial^{2}\ln P\left(\theta |x\right)}{\partial x_{i}\partial x_{j}}\right], \end{array}$$
where 1≤i, j≤3.

For the RSS-AOA time series model, the FIM is calculated using the PDF that have been given in (Equation 7). Specifically, each term has been derived as follows(11)$$\begin{array}{c} \begin{array}{cc} {\left[\mathrm{FIM}(x)\right]}_{11}=T\sum_{i=1}^{N} & \left(\frac{{\left(x_{2}-a_{i2}\right)}^{2}}{\sigma_{m_{i}}^{2}d_{2i}^{4}}+\frac{{\left(x_{1}-a_{i1}\right)}^{2}{\left(x_{3}-a_{i3}\right)}^{2}}{\sigma_{v_{i}}^{2}d_{2i}^{2}d_{i}^{4}}-\frac{\eta_{i}{\left(x_{1}-a_{i1}\right)}^{2}}{{∥x-a_{i}∥}^{4}}\right),\\ {\left[\mathrm{FIM}(x)\right]}_{22}=T\sum_{i=1}^{N} & \left(\frac{{\left(x_{1}-a_{i1}\right)}^{2}}{\sigma_{m_{i}}^{2}d_{2i}^{4}}+\frac{{\left(x_{2}-a_{i2}\right)}^{2}{\left(x_{3}-a_{i3}\right)}^{2}}{\sigma_{v_{i}}^{2}d_{2i}^{2}d_{i}^{4}}-\frac{\eta_{i}{\left(x_{2}-a_{i2}\right)}^{2}}{{∥x-a_{i}∥}^{4}}\right),\\ {\left[\mathrm{FIM}(x)\right]}_{33}=T\sum_{i=1}^{N} & \left(\frac{d_{2i}^{2}}{\sigma_{v_{i}}^{2}d_{i}^{4}}-\frac{\eta_{i}{\left(x_{3}-a_{i3}\right)}^{2}}{{∥x-a_{i}∥}^{4}}\right),\\ {\left[\mathrm{FIM}(x)\right]}_{13}=T\sum_{i=1}^{N} & \left(-\frac{\left(x_{1}-a_{i1}\right)\left(x_{3}-a_{i3}\right)}{\sigma_{v_{i}}^{2}d_{i}^{4}}-\frac{\eta_{i}\left(x_{1}-a_{i1}\right)\left(x_{3}-a_{i3}\right)}{{∥x-a_{i}∥}^{4}}\right),\\ {\left[\mathrm{FIM}(x)\right]}_{23}=T\sum_{i=1}^{N} & \left(\frac{\left(x_{2}-a_{i2}\right)\left(x_{3}-a_{i3}\right)}{\sigma_{v_{i}}^{2}d_{i}^{4}}-\frac{\eta_{i}\left(x_{2}-a_{i2}\right)\left(x_{3}-a_{i3}\right)}{{∥x-a_{i}∥}^{4}}\right),\\ {\left[\mathrm{FIM}(x)\right]}_{12}=T\sum_{i=1}^{N} & \left(-\frac{\left(x_{1}-a_{i1}\right)\left(x_{2}-a_{i2}\right)}{\sigma_{m_{i}}^{2}d_{2i}^{4}}+\frac{\left(x_{1}-a_{i1}\right)\left(x_{2}-a_{i2}\right){\left(x_{3}-a_{i3}\right)}^{2}}{\sigma_{v_{i}}^{2}d_{2i}^{2}d_{i}^{4}}-\frac{\eta_{i}\left(x_{1}-a_{i1}\right)\left(x_{2}-a_{i2}\right)}{{∥x-a_{i}∥}^{4}}\right), \end{array} \end{array}$$
where FIM(x) is a symmetric matrix, e.g., ${\left[\mathrm{FIM}(x)\right]}_{ij}={\left[\mathrm{FIM}(x)\right]}_{ji}$, i,j=1,2,3, $\eta_{i}={\left(\frac{10\gamma }{\sigma_{n_{i}}\log_{10}10}\right)}^{2}$, $d_{2i}=\sqrt{{\left(x_{1}-a_{i1}\right)}^{2}+{\left(x_{2}-a_{i2}\right)}^{2}}$, and $d_{i}=\sqrt{{\left(x_{1}-a_{i1}\right)}^{2}+{\left(x_{2}-a_{i2}\right)}^{2}+{\left(x_{3}-a_{i3}\right)}^{2}}$.

Then, the CRLB of the hybrid RSS-AOA time series model is obtained as(12)$$\begin{array}{c} \mathrm{CRLB}=\sqrt{\mathrm{trace}({\mathrm{FIM}}^{-1})}. \end{array}$$

In order to provide a more intuitive illustration of the CRLB for the RSS-AOA time series, we simulated a specific scenario.

Figure 2 illustrates a simple wireless localization scenario with randomly located target nodes and fixed anchor node locations. In this scenario, four anchor nodes are positioned at ${\left[0,10,10\right]}^{T}$, ${\left[10,30,15\right]}^{T}$, ${\left[30,10,20\right]}^{T}$, and ${\left[30,30,25\right]}^{T}$, with the noise variances of $\sigma_{m_{1}}=5$ deg, $\sigma_{v_{1}}=5$ deg, and $\sigma_{n_{1}}=1$ dB for the first anchor node, and $\sigma_{m_{i}}=10$ deg, $\sigma_{v_{i}}=10$ deg, and $\sigma_{n_{i}}=2$ dB for the rest anchor nodes, where i=2,3,4. The target nodes are randomly placed within a square with sides ranging from 1 to 40 m for $x_{1}$ and $x_{2}$, and at $x_{3}=0$. The number of time steps is set as T=5.

Figure 3 presents the CRLBs for only RSS, only AOA, and RSS-AOA time series models in the simple localization scenario. It is evident that the CRLB of RSS-AOA is the lowest. Moreover, at four specific target node locations, the FIM in (Equation 11) becomes a singular matrix without a matrix inverse, which explains the white spots in Figure 3a,c. It is worth noting that both CRLBs with AOA only and RSS only decrease as the target node approaches the center of the square region, while the CRLB of RSS-AOA reaches its minimum at the projection of the first anchor node in the square region. This is because RSS-AOA measurement only requires one node to achieve 3D localization and the first anchor node is highly accurate, resulting in lower CRLB for RSS-AOA when approaching the node, which means a lower theoretical minimum localization error.

## 3. Proposed Two-Stage Localization Method

In this section, we propose a two-stage LWLS method for solving the RSS-AOA time series localization problem with multiple types of anchor nodes. The first stage is to estimate the target node location roughly using the time-averaged RSS-AOA values. The second stage is to estimate the variance of each residual of the LWLS equation. Based on these variances, a weight matrix is calculated and used to reapply the LWLS method to obtain a more accurate solution for the target node location.

### 3.1. The First Stage

In the first stage, a rough estimation of the target location is calculated. First, the measured time series are averaged over time as ${\hat{\phi }}_{i}=\frac{1}{T}\sum_{t=1}^{T}\phi_{it}$, ${\hat{\alpha }}_{i}=\frac{1}{T}\sum_{t=1}^{T}\alpha_{it}$, ${\hat{P}}_{i}=\frac{1}{T}\sum_{t=1}^{T}P_{it}$. Using spherical coordinates, we can express the distance between the target node and the *i*-th anchor node, $∥x\;-\;a_{i}∥$, as ${\dot{u}}_{i}^{T}\left(x-a_{i}\right)$, where ${\dot{u}}_{i}$ is a unit vector in the direction of the target node from the *i*-th anchor node, which is given by(13)$$\begin{array}{c} {\dot{u}}_{i}={\left[\cos{\dot{\phi }}_{i}\sin{\dot{\alpha }}_{i},\sin{\dot{\phi }}_{i}\sin{\dot{\alpha }}_{i},\cos{\dot{\alpha }}_{i}\right]}^{T}. \end{array}$$

To simplify the process, the unit vector ${\hat{u}}_{i}$ was defined based on the average values of the sequential measurements that incorporate actual measurement noise as(14)$$\begin{array}{c} {\hat{u}}_{i}={\left[\cos{\hat{\phi }}_{i}\sin{\hat{\alpha }}_{i},\sin{\hat{\phi }}_{i}\sin{\hat{\alpha }}_{i},\cos{\hat{\alpha }}_{i}\right]}^{T}. \end{array}$$

Subsequently, the time-averaged RSS-AOA values can be expressed as follows:(15)$$\begin{array}{c} c_{i}^{T}\left(x-a_{i}\right)\approx 0, \end{array}$$(16)$$\begin{array}{c} {\left(\cos{\hat{\alpha }}_{i}{\hat{u}}_{i}-k\right)}^{T}\left(x-a_{i}\right)\approx 0, \end{array}$$(17)$$\begin{array}{c} \lambda_{i}{\hat{u}}_{i}^{T}\left(x-a_{i}\right)\approx \beta , \end{array}$$
where $c_{i}={\left[-\sin{\hat{\phi }}_{i},\cos{\hat{\phi }}_{i},0\right]}^{T}$, $k={\left[0,0,1\right]}^{T}$, $\lambda_{i}=10^{\frac{{\hat{P}}_{i}}{10\gamma }}$, $\beta =d_{0}10^{\frac{P_{0}}{10\gamma }}$. In particular, we refer to each equation in (Equation 15)–(Equation 17) as LWLS equations. Then, the target location can be obtained by applying the LWLS estimation to these 3N LWLS equations as follows:(18)$$\begin{array}{c} \begin{array}{cc} {\hat{x}}_{\mathrm{WLS}}=\arg\min_{x}\sum_{i=1}^{N}w_{i}^{2}{\left(c_{i}^{T}\left(x-a_{i}\right)\right)}^{2} & +\sum_{i=1}^{N}w_{N+i}^{2}{\left({\left(\cos{\hat{\alpha }}_{i}{\hat{u}}_{i}-k\right)}^{T}\left(x-a_{i}\right)\right)}^{2}\\ \; & +\sum_{i=1}^{N}w_{2N+i}^{2}{\left(\lambda_{i}{\hat{u}}_{i}^{T}\left(x-a_{i}\right)-\beta \right)}^{2}, \end{array} \end{array}$$
where w is a 3N×1 vector which denotes the weights of the LWLS equations. The weights in the first stage are range-based, which can be expressed as $w_{i}=w_{N+i}=w_{2N+i}=1-\frac{{\hat{d}}_{i}}{\sum_{i=1}^{N}{\hat{d}}_{i}}$, ${\hat{d}}_{i}=d_{0}10^{\frac{P_{0}-{\hat{P}}_{i}}{10\gamma }}$, i=1,2,⋯,N.

For ease of solving, (Equation 18) can be expressed as the following matrix form:(19)$$\begin{array}{c} {\hat{x}}_{\mathrm{WLS}}=\arg\min_{x}{∥W\left(Ax-b\right)∥}^{2}, \end{array}$$
where $W=\mathrm{diag}(w_{1},w_{2},\cdots ,w_{3N})$ is the weight matrix, $w_{i}$ is referred as to the weight of the *i*-th LWLS equation, and $A=\left[\begin{array}{c} \vdots \\ c_{i}^{T}\\ \vdots \\ {\left(\cos{\hat{\alpha }}_{i}{\hat{u}}_{i}-k\right)}^{T}\\ \vdots \\ \lambda_{i}{\hat{u}}_{i}^{T}\\ \vdots \end{array}\right]$ is a 3N×3 matrix, $b=\left[\begin{array}{c} \vdots \\ c_{i}^{T}a_{i}\\ \vdots \\ {\left(\cos{\hat{\alpha }}_{i}{\hat{u}}_{i}-k\right)}^{T}a_{i}\\ \vdots \\ \lambda_{i}{\hat{u}}_{i}^{T}a_{i}+\beta \\ \vdots \end{array}\right]$ is a 3N×1 vector.

Then, the closed-form solution of (Equation 19) can be obtained as follows:(20)$$\begin{array}{c} {\hat{x}}_{\mathrm{WLS}}={\left(A^{T}W^{T}WA\right)}^{-1}A^{T}W^{T}Wb. \end{array}$$

### 3.2. The Second Stage

In the second stage, the weight matrix is constructed by using the estimated variances of the LWLS equation residuals. Therefore, the variances of the residuals in (Equation 15)–(Equation 17) are first estimated.

To estimate the variance of the residual for each item in Equations (Equation 15)–(Equation 17), we can express all RSS-AOA time series measurements in the form of NT equations as follows:(21)$$\begin{array}{c} {\tilde{r}}_{1it}={\tilde{c}}_{it}^{T}\left(x-a_{i}\right), \end{array}$$(22)$$\begin{array}{c} {\tilde{r}}_{2it}={\left(\cos{\hat{\alpha }}_{i}{\tilde{u}}_{it}-k\right)}^{T}\left(x-a_{i}\right), \end{array}$$(23)$$\begin{array}{c} {\tilde{r}}_{3it}={\tilde{\lambda }}_{ij}{\tilde{u}}_{it}^{T}\left(x-a_{i}\right)-\beta , \end{array}$$
where ${\tilde{r}}_{1it}$, ${\tilde{r}}_{2it}$, and ${\tilde{r}}_{3it}$ are the residuals of (Equation 21), (Equation 22), and (Equation 23), respectively. ${\tilde{c}}_{ij}={\left[-\sin\phi_{ij},\cos\phi_{ij},0\right]}^{T}$, ${\tilde{u}}_{ij}={\left[\cos\phi_{ij}\sin\alpha_{ij},\sin\phi_{ij}\sin\alpha_{ij},\cos\alpha_{ij}\right]}^{T}$, ${\tilde{\lambda }}_{ij}=10^{\frac{P_{ij}}{10\gamma }}$.

The rough estimate of the target node location ${\hat{x}}_{\mathrm{WLS}}$ obtained in the first stage is substituted into Equations (Equation 21)–(Equation 23) to estimate the residual for each term. Then, we obtain the estimated residuals as(24)$$\begin{array}{c} {\hat{r}}_{1it}={\tilde{c}}_{it}^{T}\left({\hat{x}}_{\mathrm{WLS}}-a_{i}\right), \end{array}$$(25)$$\begin{array}{c} {\hat{r}}_{2it}={\left(\cos{\hat{\alpha }}_{i}{\tilde{u}}_{it}-k\right)}^{T}\left({\hat{x}}_{\mathrm{WLS}}-a_{i}\right), \end{array}$$(26)$$\begin{array}{c} {\hat{r}}_{3it}={\tilde{\lambda }}_{ij}{\tilde{u}}_{it}^{T}\left({\hat{x}}_{\mathrm{WLS}}-a_{i}\right)-\beta . \end{array}$$

Then, the estimated variances of the residuals in (Equation 15)–(Equation 17) are calculated as follows:(27)$$\begin{array}{c} {\hat{\sigma }}_{r_{1i}}^{2}=\frac{1}{T}\sum_{t=1}^{T}{\hat{r}}_{1it}, \end{array}$$(28)$$\begin{array}{c} {\hat{\sigma }}_{r_{2i}}^{2}=\frac{1}{T}\sum_{t=1}^{T}{\hat{r}}_{2it}, \end{array}$$(29)$$\begin{array}{c} {\hat{\sigma }}_{r_{3i}}^{2}=\frac{1}{T}\sum_{t=1}^{T}{\hat{r}}_{3it}. \end{array}$$

The LWLS estimation is used again using the residual variance-based weights, which is expressed as(30)$$\begin{array}{c} \begin{array}{cc} {\hat{x}}_{\mathrm{ML}} & =\arg\min_{x}\sum_{i=1}^{N}\frac{{\left(c_{i}^{T}\left(x-a_{i}\right)\right)}^{2}}{{\hat{\sigma }}_{r_{1i}}^{2}}+\sum_{i=1}^{N}\frac{{\left({\left(\cos{\hat{\alpha }}_{i}{\hat{u}}_{i}-k\right)}^{T}\left(x-a_{i}\right)\right)}^{2}}{{\hat{\sigma }}_{r_{2i}}^{2}}+\sum_{i=1}^{N}\frac{{\left(\lambda_{i}{\hat{u}}_{i}^{T}\left(x-a_{i}\right)-\beta \right)}^{2}}{{\hat{\sigma }}_{r_{3i}}^{2}}. \end{array} \end{array}$$

As can be seen, (Equation 30) is very similar to (Equation 18). The main difference is the weights in the first stage are range-based, while the weights in the second stage are variance-based, where $w_{i}^{2}=\frac{1}{\sigma_{r_{1i}}^{2}}$, $w_{N+i}^{2}=\frac{1}{\sigma_{r_{2i}}^{2}}$, and $w_{2N+i}^{2}=\frac{1}{\sigma_{r_{3i}}^{2}}$.

The matrix form of the ML estimation is written as(31)$$\begin{array}{c} {\hat{x}}_{\mathrm{ML}}=\arg\min_{x}{∥\tilde{W}\left(Ax-b\right)∥}^{2}, \end{array}$$
where $\tilde{W}=\mathrm{diag}\left(\frac{1}{{\hat{\sigma }}_{r_{11}}},\frac{1}{{\hat{\sigma }}_{r_{12}}},\cdots ,\frac{1}{{\hat{\sigma }}_{r_{1N}}},\frac{1}{{\hat{\sigma }}_{r_{21}}},\frac{1}{{\hat{\sigma }}_{r_{22}}},\cdots ,\frac{1}{{\hat{\sigma }}_{r_{2N}}},\frac{1}{{\hat{\sigma }}_{r_{31}}},\frac{1}{{\hat{\sigma }}_{r_{32}}},\cdots ,\frac{1}{{\hat{\sigma }}_{r_{3N}}}\right)$.

Finally, the closed-form solution of the estimated target localization in the second stage is obtained as(32)$$\begin{array}{c} {\hat{x}}_{\mathrm{WLS}2}={\left(A^{T}{\tilde{W}}^{T}\tilde{W}A\right)}^{-1}A^{T}{\tilde{W}}^{T}\tilde{W}b. \end{array}$$

It is worth noting that the proposed algorithm can combine AOA with any signal propagation model that provides distance measurements, including ultra-wideband ranging. To adapt the algorithm, Equation (Equation 17) should be modified to ${\hat{u}}_{i}^{T}\left(x-a_{i}\right)\approx d_{i}$, where $d_{i}$ is the measured distance. Additionally, the noise modeling for Equation (Equation 17) should be updated accordingly.

## 4. Simulation Results

In this section, simulations are conducted to demonstrate the performance of the proposed localization method. The simulations can be considered as locating a smartphone (target node) using routers (anchor nodes), where the AOA can be calculated using the MUSIC algorithm with the router’s antennas, and the RSS can be obtained directly from the communication protocol, for example, Wi-Fi (IEEE 802.11) or Bluetooth (IEEE 802.15.1). The performance of the proposed method is compared with existing state-of-the-art methods, including LS [11], WLS-d [12], ECWLS [13], TSLS [14], ENWLS [15], SDP-SOCP [18], and GTRS [19]. These methods do not use time series measurements, so the average values of time series measurements over time are taken as the input data for the compared methods. It is worth noting that the average measurement values over time are more accurate than the measurement values at any given time, and the variance of the measurement noise becomes $\frac{1}{T}$ of the measurement noise variance at any given time. The CRLB is also included in each simulation to provide a theoretical benchmark for comparison with the actual performance of the considered localization methods.

In all simulations, the anchor and target nodes were randomly distributed within a cubic space with a side length of 40 m. *M* independent Monte Carlo (Mc) runs were performed, where the locations of both the anchor and target nodes were re-randomized in each run. The measurement noise for each anchor node followed a zero-mean Gaussian distribution. Note that the variance of each measurement noise is not identical. The standard deviation of the measurement noise for RSS, azimuth angle, and elevation angle followed an exponential distribution with mean values of $\mu_{n}$, $\mu_{m}$, and $\mu_{v}$, respectively. The performance is assessed through the calculation of root mean square error (RMSE), which is computed by(33)$$\begin{array}{c} \mathrm{RMSE}=\sqrt{\frac{1}{M}\sum_{i=1}^{M}{∥x_{i}-{\hat{x}}_{i}∥}^{2}}, \end{array}$$
where $x_{i}$ and ${\hat{x}}_{i}$ are the true location and the estimated location, respectively, of the target node in the *i*-th Mc run. In this paper, we set M=3000 for all simulations. Based on common wireless communication environments and empirical experience, the reference received signal power, $P_{0}$, is set to −10 dBm, and the reference distance, $d_{0}$, is set to 1 m. The path-loss exponent, γ, represents the rate at which the signal attenuates with increasing distance. In open spaces, γ is typically 2. However, in complex environments, γ can be larger, usually ranging between 2 and 6. Without loss of generality, this paper selects γ=2.7. In the comparison, we vary the number of anchor nodes *N*, the length of the measurement sequence *T*, and the distribution variables $\mu_{n}$, $\mu_{m}$, and $\mu_{v}$. These values will be introduced in each scenario. The MATLAB toolbox CVX [24] is utilized to solve the SDP and SOCP, with SeDuMi [25] as the selected solver.

The computational complexity of the proposed method and the compared methods are first presented, followed by the comparison between the proposed method and the state-of-the-art methods.

### 4.1. Computational Complexity

Table 1 lists the complexity of all the considered methods. *N* and *T* denote the numbers of the anchor nodes and the time steps. *K* represents the number of iterations in the bisection method used in solving the GTRS problem. As shown in this table, the complexity of LS, WLS-d, ECWLS, TSLS, and ENWLS is linear, while the proposed algorithm has the largest complexity among the LWLS methods. This is because the proposed method requires TN iteration calculations for computing the weight matrix, while the other methods only require *N* iteration calculations. But this does not affect the real-time implementation of the proposed method, as will be demonstrated in the average runtime table. The SDP-SOCP method has the highest computational complexity among the compared methods, which may hinder its real-time implementation.

To provide a more intuitive understanding, the average runtime for the considered method is also investigated in Table 2, which was obtained using a MacBook with an M1 chip. Clearly, the results in Table 2 support the key findings about the computational complexity of the considered methods in Table 1. Although the proposed method has higher complexity and longer runtime than the compared LWLS methods, it can still achieve real-time localization. This is because its runtime is very low in practice (less than 1 ms), and it scales linearly with the number of anchor nodes and time steps, making it suitable for real-time applications.

### 4.2. Comparison with the State-of-the-Art Methods

#### 4.2.1. Scenario 1

In this scenario, we examine the localization performance of different methods by varying the mean standard deviation, $\mu_{m}$, of the azimuth measurement errors. Figure 4 shows the comparison results when N=5, T=5, $\mu_{v}=10$ deg, and $\mu_{n}=6$ dB. It is evident that the proposed method exhibits an absolute advantage among the compared methods, owing to its analysis of the measurement errors of each node and the calculation of reasonable weights for weighted least squares. Therefore, it performs well in this scenario.

#### 4.2.2. Scenario 2

In this scenario, we investigate the impact of varying the mean standard deviation, $\mu_{v}$, of the measurement errors in elevation angle. Figure 5 displays the comparison results when N=5, T=5, $\mu_{m}=10$ deg, and $\mu_{n}=6$ dB. The proposed method stands out from other algorithms, as it accounts for the measurement errors at each node and employs a weighted least squares approach with appropriate weights. This results in superior performance in the considered scenario, validating the effectiveness of the proposed approach.

#### 4.2.3. Scenario 3

In this scenario, we examine the localization performance of different methods by varying the mean standard deviation, $\mu_{n}$, of the RSS measurement errors. Figure 6 presents the comparison results when N=5, T=5, $\mu_{m}=10$ deg, and $\mu_{v}=10$ deg. From the figure, it can be observed that the RMSEs of the TSLS, WLS-d, and ENWLS are insensitive to changes in $\mu_{n}$, while the LS exhibits a significant increase with the increase in $\mu_{n}$. Different weight calculations lead to different outcomes, even though they are all WLS methods. The RMSE of the CRLB increases slightly as $\mu_{n}$ increases, which means that the theoretical lower bound of the localization error also increases with $\mu_{n}$. This implies that the weights of RSS measurement are either too small (in TSLS, WLS-d, and ENWLS methods) or too large (in LS methods). The RMSE of the proposed method increases slightly as $\mu_{n}$ increases, which is consistent with the trend of the CRLB. This shows that the weights calculated by the proposed method are reasonable.

#### 4.2.4. Scenario 4

In this scenario, we set the parameters of the measurement error distributions as $\mu_{m}=6$ deg, $\mu_{v}=6$ deg, and $\mu_{n}=4$ dB, and the time steps of the sequential measurement as T=5. Figure 7 shows the RMSEs versus the number of anchor nodes, *N*. From this figure, we can see that the RMSEs of all the considered methods decrease when *N* increases. Furthermore, one can see that more anchors lead to better performance for all algorithms, as they provide more reliable information. It also demonstrates the advantage of using combined measurements in hybrid systems over using single measurements in traditional systems.

#### 4.2.5. Scenario 5

In this scenario, we keep the number of anchor nodes fixed and vary the time steps of the measurements. Specifically, we fix the number of anchor bodes to N=10, while varying the number of time steps, *T* from 3 to 10, with a fixed parameters of the measurement error distributions are $\mu_{m}=6$ deg, $\mu_{v}=6$ deg, and $\mu_{n}=4$ dB. The results in Figure 8 show that the performance of the considered method improves as the time steps increase. We can see from the figure that the proposed method has low localization accuracy when T=3 and T=4, because there are not enough data points to estimate the standard deviation of measurement errors at each anchor node. This leads to significant errors in the weight calculations and consequently, large positioning errors. However, when T≥5, the proposed method achieves high localization accuracy, with RMSE values close to the CRLB.

## 5. Conclusions and Future Work

This paper presents a novel localization approach that uses hybrid RSS-AOA time series measurements and considers the varying measurement noise distributions among different types of anchor nodes. This improves the positioning accuracy and reliability in heterogeneous anchor node scenarios. We propose a two-stage LWLS method that uses weights based on the residual variances to handle different types of anchor nodes.The proposed method consists of two stages. In the first stage, we use the range-based LWLS method to estimate a rough target location. In the second stage, we estimate the residual variances of LWLS equations and use them as the LWLS weights to reapply the LWLS method and obtain a more accurate location. Simulation experiments show the superior performance of the proposed method. It achieves more accurate and reliable localization results in heterogeneous anchor node scenarios than existing methods. Moreover, the proposed method does not increase the complexity of the localization system significantly, making it practical for real applications.

The simulation results show that using the residual variances of LWLS equations as weights for localization is an effective approach in heterogeneous anchor node scenarios. Our future work will focus on three main aspects: (1) Improving the accuracy of estimating the residual variances of LWLS equations to further enhance the localization performance. (2) Developing noise models for the unit vectors in spherical coordinates to better reflect real-world conditions. (3) Reducing the number of time steps required in the time series, making the proposed method more efficient and adaptable to various application scenarios. We will also investigate the applicability of the proposed method in real-world application scenarios and evaluate its performance in practical environments.

## Figures and Tables

**Figure 1 sensors-25-02028-f001:**
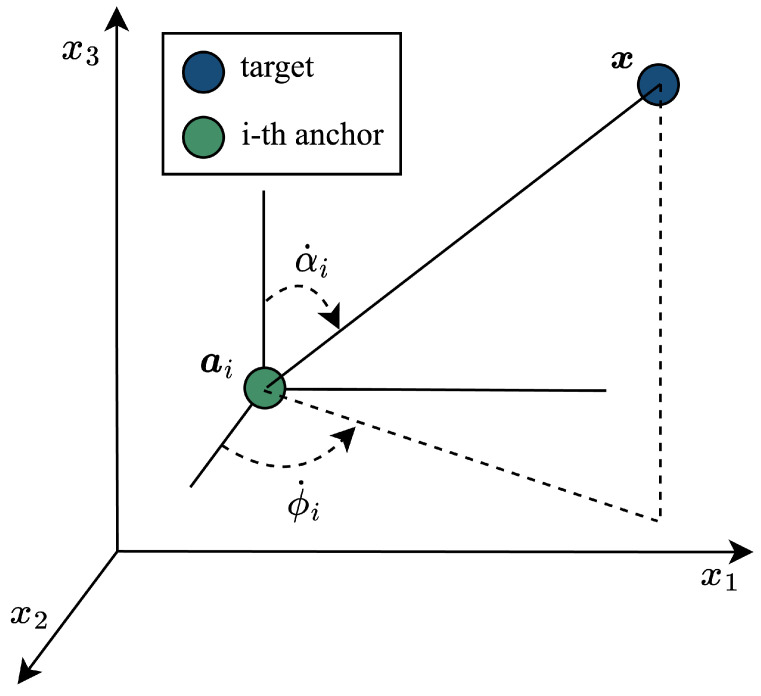
Illustration of an anchor node and the target node in 3D space.

**Figure 2 sensors-25-02028-f002:**
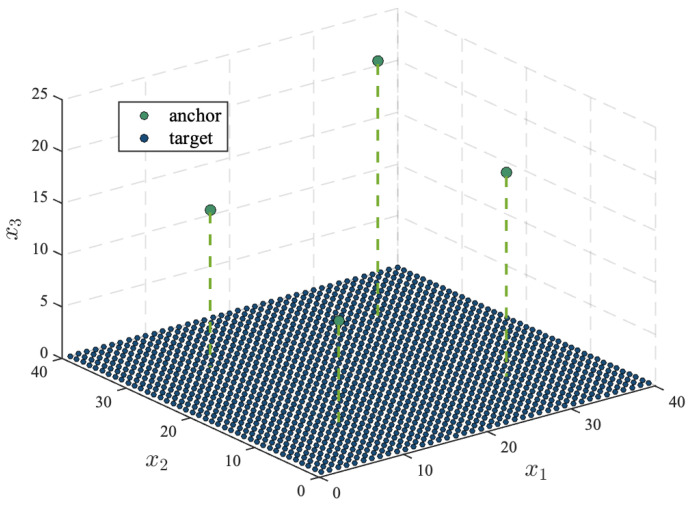
Node placement of a simple localization scenario.

**Figure 3 sensors-25-02028-f003:**
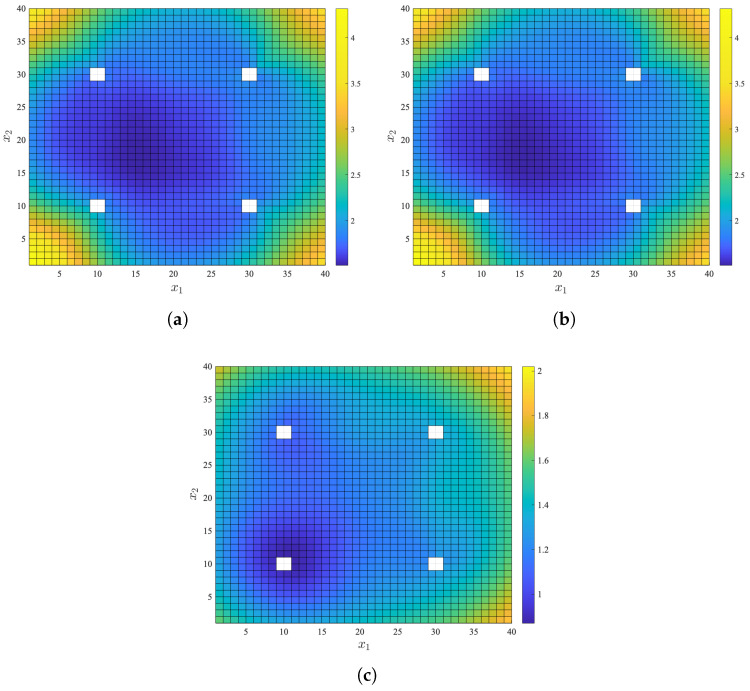
The CRLBs for the simple localization scenario. (**a**) CRLB for only AOA time series. (**b**) CRLB for only RSS time series. (**c**) CRLB for RSS-AOA time series.

**Figure 4 sensors-25-02028-f004:**
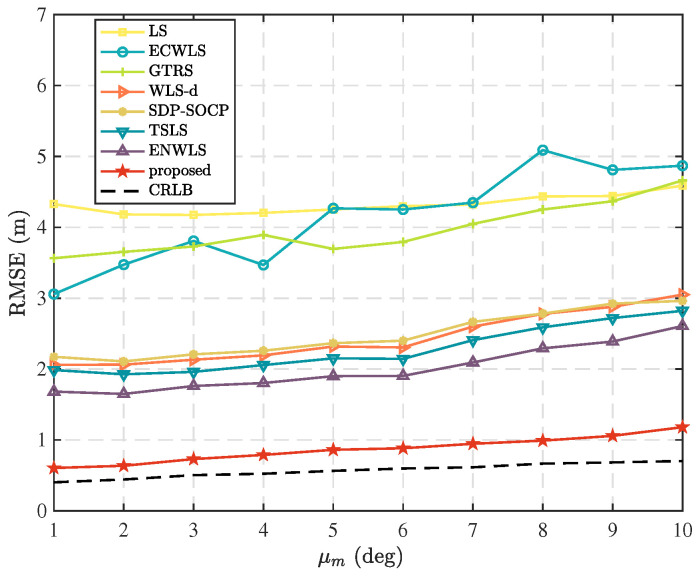
Comparison of RMSE as mean standard deviation, $\mu_{m}$, of azimuth measurement error distribution.

**Figure 5 sensors-25-02028-f005:**
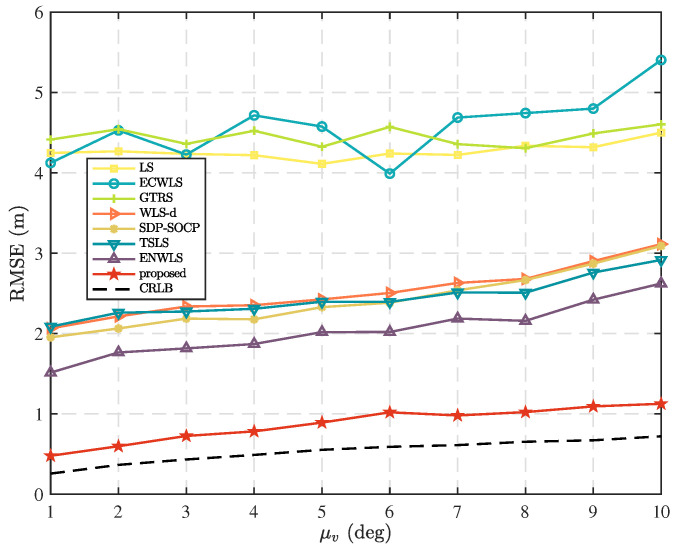
Comparison of RMSE as mean standard deviation, $\mu_{v}$, of elevation measurement error distribution.

**Figure 6 sensors-25-02028-f006:**
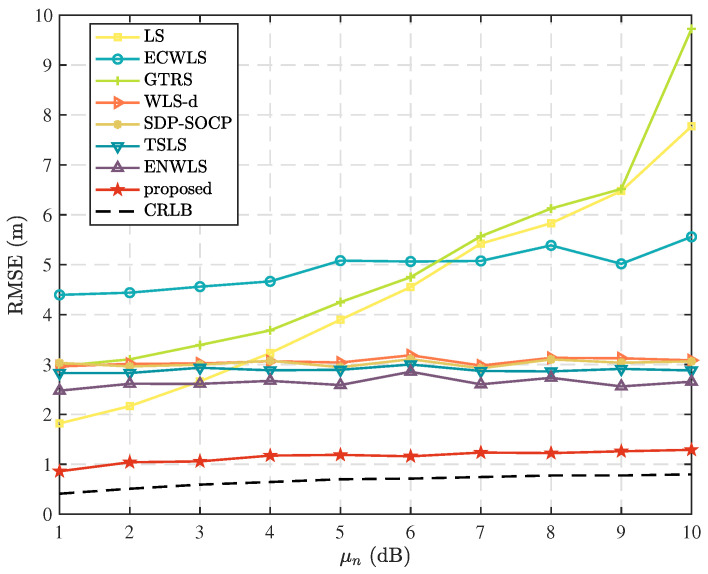
Comparison of RMSE as mean standard deviation, $\mu_{n}$, of RSS measurement error distribution.

**Figure 7 sensors-25-02028-f007:**
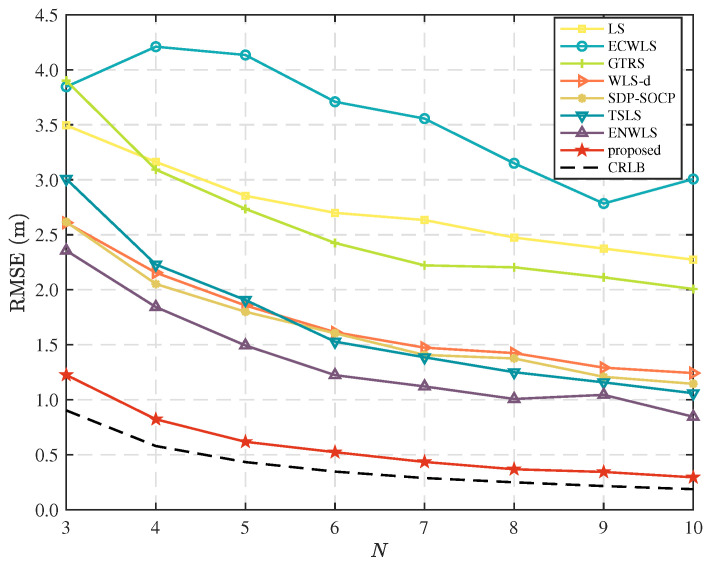
Comparison of RMSE as the number of anchor nodes, *N*.

**Figure 8 sensors-25-02028-f008:**
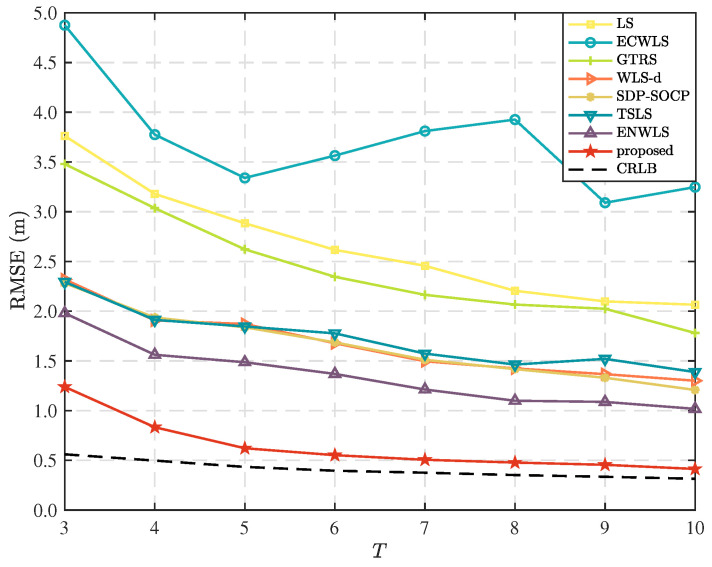
Comparison of RMSE as time steps, *T*.

**Table 1 sensors-25-02028-t001:** The complexity of the considered localization methods.

Algorithm	Complexity
the proposed algorithm	O(TN)
LS in [11]	O(N)
WLS-d in [12]	O(N)
ECWLS in [13]	O(N)
TSLS in [14]	O(N)
ENWLS in [15]	O(N)
SDP-SOCP in [18]	$O(N^{3.5})$
GTRS in [19]	O(KN)

**Table 2 sensors-25-02028-t002:** The average runtime of the considered algorithms.

Algorithm	Time (s)
The proposed algorithm	$7.46\times 10^{-4}$
LS in [11]	$7.00\times 10^{-5}$
WLS-d in [12]	$9.13\times 10^{-5}$
ECWLS in [13]	$2.64\times 10^{-4}$
TSLS in [14]	$1.80\times 10^{-4}$
ENWLS in [15]	$3.03\times 10^{-4}$
SDP-SOCP in [18]	$7.65\times 10^{-1}$
GTRS in [19]	$7.40\times 10^{-4}$

## Data Availability

No new data were created or analyzed in this study. Data sharing is not applicable to this article.

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
