# Peer review of "A Simple and Efficient Method for RSS-AOA-Based Localization with Heterogeneous Anchor Nodes"

_sensors, 2025, doi:10.3390/s25072028_

Round 1
Reviewer 1 Report
Comments and Suggestions for Authors
Paper is not acceptable in its current form. Authors should be encouraged to resubmit a rewritten version after the changes suggested in the Comments Section.
The proposed approach uses two sequential stages. Initially, a range-based LWLS method is applied to obtain a preliminary rough estimate of the target’s location. In the subsequent stage, the residual variances from the LWLS equations are obtained and utilized as weights to refine the LWLS estimation, “leading” to a more precise location. The simulation results confirm that the proposed method outperforms alternative approaches.
Comments Section:
1.- How do you explain that rough position estimates obtained from Eq. (32) with large errors, in the first stage, can help to provide more precise locations in the second stage using their variance residuals? Also, the result section shows that the proposed scheme overcomes well-known algorithms in all tests. Moreover, in future work, authors claim that improving the accuracy of estimating the residual variance, obtained in the first stage, can improve even more the localization performance.
2.- Can you elaborate on how you obtain or deduce the values of the path-loss factor, Po, and d0 (lines 271-272)?
3.- In Table 1. How do you deduce the computational complexity of each algorithm?
4.- Why do you argue that your algorithm can run real-time applications in lines 291-292?
Author Response
Original Manuscript ID: sensors-3514454
Original Article Title: “A Simple and Efficient Method for RSS-AOA Based Localization with
Heterogeneous Anchor Nodes”
To: sensors Editor
Re: Response to reviewers
Dear Editor and Reviewers,
Thank you very much for taking the time to review our manuscript. We greatly appreciate all your insightful comments and suggestions. We have thoroughly reviewed and addressed each point, and have revised our manuscript accordingly. The modifications within the manuscript have been highlighted in yellow for easier reference.
We are uploading the following documents:
(a) our point-by-point response to the comments (below) also known as our 'response to reviewers'.
(b) an updated version of our manuscript with changes indicated in yellow highlighting.
(c) a compressed package containing a LaTeX file (ZIP main document).
We hope that our revisions have adequately addressed your concerns, and we look forward to your feedback on the revised manuscript.
Best regards,
Weizhong Ding, Lincan Li, and Shengming Chang.
Comment 1: How do you explain that rough position estimates obtained from Eq. (32) with large errors, in the first stage, can help to provide more precise locations in the second stage using their variance residuals? Also, the result section shows that the proposed scheme overcomes well-known algorithms in all tests. Moreover, in future work, authors claim that improving the accuracy of estimating the residual variance, obtained in the first stage, can improve even more the localization performance.
Response: Thank you for your question. The errors in the rough position estimates from the first stage are large compared to the final estimates of the proposed method. However, the variance residuals based on these rough estimates still reflect the precision of the corresponding anchor nodes. By using these residuals to determine the weights of each anchor node in the second stage, the algorithm can avoid the situation where large errors from certain nodes significantly degrade the accuracy of the final estimate.
Comment 2: Can you elaborate on how you obtain or deduce the values of the path-loss factor, Po, and d0 (lines 271-272)?
Response: Thank you for your question regarding the values of the path-loss factor (γ), reference received signal power (P0), and reference distance (d0). These values were selected based on common practices in wireless communication environments and empirical experience, making them suitable for indoor localization scenarios. We have added the following explanation to the manuscript: "Based on common wireless communication environments and empirical experience, the reference received signal power, P0, is set to −10 dBm, the reference distance, d0​, is set to 1 m. The path-loss exponent, γ, represents the rate at which the signal attenuates with increasing distance. In open areas, γ is typically 2. However, in complex environments (e.g., indoor or obstructed areas), γ can be larger, usually ranging between 2 and 6. Without loss of generality, this paper selects γ=2.7.”
Comment 3: In Table 1. How do you deduce the computational complexity of each algorithm?
Response: Thank you for your question. In Table 1, the computational complexities of all algorithms except the proposed one are directly obtained from their corresponding references. The complexities of the linear algorithms are clearly O(N). For GTRS, the bisection method is used, resulting in a complexity of O(KN). For SDP-SOCP, the interior-point method and matrix decomposition are employed, leading to a complexity of O(N^3.5). Detailed proofs of these complexities can be found in the corresponding references. The complexity of the proposed algorithm is explained in Section 4.1 of our paper.
Comment 4: Why do you argue that your algorithm can run real-time applications in lines 291-292?
Response: Thank you very much for your question. We have added the following explanation to the manuscript: "This is because its runtime is very low in practice (less than 1 millisecond), and it scales linearly with the number of anchor nodes and time steps, making it suitable for real-time applications."

Reviewer 2 Report
Comments and Suggestions for Authors
In general, the problem statement in the article is motivated and practically justified for such a well-developed problem as localization relative to beacons, on the subject of which there are many publications. My comments are mainly related to the practical aspects of choosing the model parameters for checking its accuracy and testing. After all, indoor positioning is a practical thing, and I think that a completely abstract choice of model scenarios here is not advisable.
- For what reasons were the parameters chosen for modeling: the dispersion of the estimate of the angle of arrival of the received signal and the dispersion of its power? How do the model parameters of the noise component level of the angle of arrival and the received power relate to the real spread of the measured angles and measured signal powers? What is the reason for choosing the attenuation index gamma - 2.7? A value greater than two most likely corresponds to a situation where there is no direct beam, and this means a fairly large spread of the measurement of the signal power and its angle of arrival. Is this what the authors are interested in? The attenuation index for a direct beam in line-of-sight conditions will be less than two.
- Why didn't the authors choose a realistic indoor signal propagation model when selecting the model parameters, of which there are quite a few for both narrowband and ultra-wideband wireless systems? Using such models in the simulation would make the results significantly more valuable and specific for wireless system specialists.
- The literature review could have been done more, specifically taking into account the works that provide data on the actual spread of signal arrival angles and signal power values ​​that are encountered in practice. These values ​​depend greatly on the propagation conditions, and the measured signal strength indicator is generally unpredictable. For narrowband systems it is one thing, and for UWB it will be completely different. Did the authors make any assumptions on this score? After all, positioning is a very practical thing, and it is a bit strange to consider the problem without such data.
- It is not very clear what the point is in Table 2, which shows The average runtime of the considered algorithms. I assume There will be different values ​​on another computer, and still others on the hardware platform of a real device.
- The authors focus on the numerical aspects of assessing the accuracy of determining coordinates. In this regard, an additional question arises: what kind of arithmetic does the process of assessing coordinates require, in the authors' opinion? Ten-byte doubles? Are all calculations performed on a stationary computer offline? It would be good to clarify these points in the text of the work. What is the intended scenario for using the author's method in reality? Or do the authors position the presented material as a first step that reveals only the fundamental points and requires revision for use in real devices?
- It would be good to clearly indicate in the text which practical scenarios in the operation of wireless networks correspond to Scenarios 1-5.
In general, the work is interesting as an approach to processing raw data obtained from real wireless devices.
Author Response
Original Manuscript ID: sensors-3514454
Original Article Title: “A Simple and Efficient Method for RSS-AOA Based Localization with
Heterogeneous Anchor Nodes”
To: sensors Editor
Re: Response to reviewers
Dear Editor and Reviewers,
Thank you very much for taking the time to review our manuscript. We greatly appreciate all your insightful comments and suggestions. We have thoroughly reviewed and addressed each point, and have revised our manuscript accordingly. The modifications within the manuscript have been highlighted in yellow for easier reference.
We are uploading the following documents:
(a) our point-by-point response to the comments (below) also known as our 'response to reviewers'.
(b) an updated version of our manuscript with changes indicated in yellow highlighting.
(c) a compressed package containing a LaTeX file (ZIP main document).
We hope that our revisions have adequately addressed your concerns, and we look forward to your feedback on the revised manuscript.
Best regards,
Weizhong Ding, Lincan Li, and Shengming Chang.
Comment 1: For what reasons were the parameters chosen for modeling: the dispersion of the estimate of the angle of arrival of the received signal and the dispersion of its power? How do the model parameters of the noise component level of the angle of arrival and the received power relate to the real spread of the measured angles and measured signal powers? What is the reason for choosing the attenuation index gamma - 2.7? A value greater than two most likely corresponds to a situation where there is no direct beam, and this means a fairly large spread of the measurement of the signal power and its angle of arrival. Is this what the authors are interested in? The attenuation index for a direct beam in line-of-sight conditions will be less than two.
Response: Thank you very much for your question. The angle of arrival (AOA) was chosen as a system parameter primarily because of its high precision. Signal power, on the other hand, is low-cost and can be effectively combined with AOA to generate a closed-form solution for the localization problem. Regarding noise, we assume that the noise in AOA and received power is completely independent within the same anchor node and follows different Gaussian distributions. Additionally, the noise across different anchor nodes is also independent and follows different Gaussian distributions.
The value of γ is set to 2.7 because we consider a more general scenario (including both LOS and NLOS conditions). In fact, many studies have chosen values greater than 2 for γ, such as 2.2–2.8 in [1] and 4 in [2]. We have added the following explanation to the manuscript: "The path-loss exponent, γ, represents the rate at which the signal attenuates with increasing distance. In open areas, γ is typically 2. However, in complex environments (e.g., indoor or obstructed areas), γ can be larger, usually ranging between 2 and 6. Without loss of generality, this paper selects γ=2.7."
References:
[1] F. Watanabe, “Wireless sensor network localization using AoA measurements with two-step error variance-weighted least squares,” IEEE Access, vol. 9, pp. 10820–10828, 2021, doi: 10.1109/ACCESS.2021.3050309.
[2] S. Chang, Y. Zheng, P. An, J. Bao, and J. Li, “3-D RSS-AOA based target localization method in wireless sensor networks using convex relaxation,” IEEE Access, vol. 8, pp. 106901–106909 2020, doi: 10.1109/ACCESS.2020.3000793.
Comment 2: Why didn't the authors choose a realistic indoor signal propagation model when selecting the model parameters, of which there are quite a few for both narrowband and ultra-wideband wireless systems? Using such models in the simulation would make the results significantly more valuable and specific for wireless system specialists.
Response: Thank you for your valuable suggestion. The primary reason for not using a more realistic indoor signal propagation model in this study is to maintain a balance between generality and simplicity. In fact, any model that can obtain the distance between anchor nodes and the target node can be used with the proposed algorithm. We have added an explanation in the manuscript on how to use distance instead of received signal strength in the proposed method.
Comment 3: The literature review could have been done more, specifically taking into account the works that provide data on the actual spread of signal arrival angles and signal power values ​​that are encountered in practice. These values ​​depend greatly on the propagation conditions, and the measured signal strength indicator is generally unpredictable. For narrowband systems it is one thing, and for UWB it will be completely different. Did the authors make any assumptions on this score? After all, positioning is a very practical thing, and it is a bit strange to consider the problem without such data.
Response: Thank you very much for your valuable suggestion. Indeed, considering studies that provide data on the actual spread of signal arrival angles and signal power values would be beneficial for engineering practice. However, the primary focus of this paper is on the backend algorithm implementation and simulation validation of the algorithm's effectiveness. In future work, we will conduct research in this direction. Thank you again for your valuable feedback
Comment 4: It is not very clear what the point is in Table 2, which shows The average runtime of the considered algorithms. I assume There will be different values ​​on another computer, and still others on the hardware platform of a real device.
Response: Thank you for your careful reading. We agree that the runtime values in Table 2 may vary on different computers or hardware platforms. To provide clarity, we have added the explanation to the manuscript. (line 303)
Comment 5: The authors focus on the numerical aspects of assessing the accuracy of determining coordinates. In this regard, an additional question arises: what kind of arithmetic does the process of assessing coordinates require, in the authors' opinion? Ten-byte doubles? Are all calculations performed on a stationary computer offline? It would be good to clarify these points in the text of the work. What is the intended scenario for using the author's method in reality? Or do the authors position the presented material as a first step that reveals only the fundamental points and requires revision for use in real devices?
Response: Thank you for your questions. The proposed method uses double-precision floating-point arithmetic (64-bit doubles), which ensures numerical stability and accuracy. All simulations were performed offline on a stationary computer (MacBook with an M1 chip). The method is designed for real-world scenarios like indoor localization in smart buildings or IoT applications, where heterogeneous anchor nodes with varying noise distributions are present. While the current work focuses on fundamental aspects, further optimizations may be needed for deployment on real devices.
Comment 6: It would be good to clearly indicate in the text which practical scenarios in the operation of wireless networks correspond to Scenarios 1-5.
Response: Thank you very much for your suggestion. We have added a description of the practical environment to the manuscript: "The simulations can be considered as locating a smartphone (target node) using routers (anchor nodes), where the AOA can be calculated using the MUSIC algorithm with the router's antennas, and the RSS can be obtained directly from the communication protocol, for example, Wi-Fi (IEEE 802.11) or Bluetooth (IEEE 802.15.1)."

Reviewer 3 Report
Comments and Suggestions for Authors
The article introduces a novel localization approach for wireless sensor networks, leveraging a hybrid time-series measurement technique that integrates received signal strength and angle of arrival to enhance positioning accuracy and reliability in environments with heterogeneous anchor nodes. The proposed method is evaluated through simulations across diverse scenarios, systematically varying measurement errors and time steps to assess its robustness and effectiveness.
The paper is well-written and presents a well-developed model grounded in statistical distributions and mathematical formulations. The simulation scenarios effectively capture all key variations in model parameters, ensuring a comprehensive evaluation of the proposed approach.
Equations (4), (5), and (6) introduce noise components—azimuth angle, elevation angle, and received signal strength—at each anchor node i across multiple time instances t. The cumulative effect of these noise variations directly impacts the complexity of the proposed method, influencing both computational load and localization accuracy. This complexity necessitates adaptive noise modeling and efficient estimation strategies to ensure reliable positioning despite measurement fluctuations.
In line 160, the authors assume that the standard deviations associated with different anchor nodes follow exponential distributions. The realism of this assumption depends on the underlying characteristics of the sensor network environment. Exponential distributions are often used to model noise and uncertainty in wireless communications, particularly in cases where variations are memoryless and exhibit a rapid decay in probability as deviations increase. However, in practical scenarios, noise sources such as multipath fading, environmental obstructions, and hardware inconsistencies may follow more complex distributions, such as Gaussian models. Empirical derivation of the distribution is also possible but challenging due to the requirement of extensive data collection and calibration.
The authors claim to use heterogeneous anchors, yet the study lacks evidence of variations in node types, transmission power, or measurement accuracy, leaving the claim unsubstantiated.
The simulation does not assess the scalability of the proposed method. To provide a comprehensive evaluation, the authors should vary both the number of anchor nodes and sensor nodes, analyzing their impact on localization accuracy and computational efficiency.
Given that the localization approach operates in 3D, the authors should also evaluate scenarios where sensor node positions are non-uniform and not confined to the same plane, as depicted in Figure 2.
The study does not consider scenarios where a sensor node fails to reach all three anchors, which is a critical limitation in real-world deployments. Incorporating hybrid optimization techniques, such as those explored in [10.1109/ACCESS.2024.3417227], could enhance robustness and adaptability in cases of limited anchor connectivity.
Similarly, the study does not explore scenarios where a sensor node utilizes more than three anchors to mitigate noise effects and improve localization accuracy. Incorporating additional anchor data could enhance robustness against measurement errors and improve overall positioning reliability.
Author Response
Original Manuscript ID: sensors-3514454
Original Article Title: “A Simple and Efficient Method for RSS-AOA Based Localization with
Heterogeneous Anchor Nodes”
To: sensors Editor
Re: Response to reviewers
Dear Editor and Reviewers,
Thank you very much for taking the time to review our manuscript. We greatly appreciate all your insightful comments and suggestions. We have thoroughly reviewed and addressed each point, and have revised our manuscript accordingly. The modifications within the manuscript have been highlighted in yellow for easier reference.
We are uploading the following documents:
(a) our point-by-point response to the comments (below) also known as our 'response to reviewers'.
(b) an updated version of our manuscript with changes indicated in yellow highlighting.
(c) a compressed package containing a LaTeX file (ZIP main document).
We hope that our revisions have adequately addressed your concerns, and we look forward to your feedback on the revised manuscript.
Best regards,
Weizhong Ding, Lincan Li, and Shengming Chang.
Comment 1: Equations (4), (5), and (6) introduce noise components—azimuth angle, elevation angle, and received signal strength—at each anchor node i across multiple time instances t. The cumulative effect of these noise variations directly impacts the complexity of the proposed method, influencing both computational load and localization accuracy. This complexity necessitates adaptive noise modeling and efficient estimation strategies to ensure reliable positioning despite measurement fluctuations.
Response: Thank you for your comment. We model all noise components (azimuth angle, elevation angle, and received signal strength) as Gaussian-distributed noise. In the second stage of the algorithm, these Gaussian-distributed noise components are estimated. Specifically, the noise distribution for each measurement parameter at each node is assumed to be different and is estimated separately. This approach allows the algorithm to adapt to the varying noise characteristics of different anchor nodes, ensuring reliable positioning despite measurement fluctuations.
Comment 2: In line 160, the authors assume that the standard deviations associated with different anchor nodes follow exponential distributions. The realism of this assumption depends on the underlying characteristics of the sensor network environment. Exponential distributions are often used to model noise and uncertainty in wireless communications, particularly in cases where variations are memoryless and exhibit a rapid decay in probability as deviations increase. However, in practical scenarios, noise sources such as multipath fading, environmental obstructions, and hardware inconsistencies may follow more complex distributions, such as Gaussian models. Empirical derivation of the distribution is also possible but challenging due to the requirement of extensive data collection and calibration.
Response: Thank you very much for your comment. Indeed, modeling the standard deviations of different anchor nodes in real-world localization scenarios is challenging, and Gaussian models are a possible and reasonable approach. However, in this paper, we chose to use exponential distributions to model the standard deviations, as this allows for a greater variation in the precision of different anchor nodes (including the possibility of extremely imprecise nodes). We have added an explanation of this choice to the manuscript. Thank you again for your valuable feedback. (lines 165-169)
Comment 3: The authors claim to use heterogeneous anchors, yet the study lacks evidence of variations in node types, transmission power, or measurement accuracy, leaving the claim unsubstantiated.
Response: Thank you very much for your question. Indeed, for different types of anchors, the transmission power may vary. To maintain a balance between generality and simplicity, we assumed that the reference received signal power (P0) is the same for all anchors, but the measurement noise (precision) varies among them.
Comment 4: The simulation does not assess the scalability of the proposed method. To provide a comprehensive evaluation, the authors should vary both the number of anchor nodes and sensor nodes, analyzing their impact on localization accuracy and computational efficiency.
Response: Thank you very much for your suggestion. In Scenario 3 of the simulation, we conducted experiments to analyze the impact of the number of anchor nodes on localization accuracy.
Comment 5: Given that the localization approach operates in 3D, the authors should also evaluate scenarios where sensor node positions are non-uniform and not confined to the same plane, as depicted in Figure 2.
Response: Thank you very much for your suggestion. In all simulations, the anchor and target nodes were randomly distributed within a cubic space with a side length of 40 m. Therefore, the sensor node positions are indeed not confined to the same plane.
Comment 6: The study does not consider scenarios where a sensor node fails to reach all three anchors, which is a critical limitation in real-world deployments. Incorporating hybrid optimization techniques, such as those explored in [10.1109/ACCESS.2024.3417227], could enhance robustness and adaptability in cases of limited anchor connectivity.
Response: Thank you for your valuable feedback. In this study, we primarily focus on addressing the issue of varying measurement accuracies among different anchor nodes. When the number of accessible anchor nodes is fewer than three, the WLS method can still be directly applied. In a hybrid RSS-AOA environment, only one anchor node is required to achieve localization.
Comment 7: Similarly, the study does not explore scenarios where a sensor node utilizes more than three anchors to mitigate noise effects and improve localization accuracy. Incorporating additional anchor data could enhance robustness against measurement errors and improve overall positioning reliability.
Response: Thank you very much for your suggestion. In the simulation experiments, specifically in Scenario 3, we tested the impact of using more than three anchor nodes on the localization performance (as shown in Figure 7). Indeed, increasing the number of anchor nodes can improve both the accuracy and reliability of localization.

Round 2
Reviewer 1 Report
Comments and Suggestions for Authors
The manuscript's authors have thoroughly responded to all the raised questions. In my view, the article meets the standards for publication in this prestigious journal.
Reviewer 2 Report
Comments and Suggestions for Authors
I have no more comments.